# Development and Optimisation of Inhalable EGCG Nano-Liposomes as a Potential Treatment for Pulmonary Arterial Hypertension by Implementation of the Design of Experiments Approach

**DOI:** 10.3390/pharmaceutics15020539

**Published:** 2023-02-06

**Authors:** Fatma Haddad, Nura Mohammed, R. C. Gopalan, Yousef Al Ayoub, Md Talat Nasim, K. H. Assi

**Affiliations:** 1School of Pharmacy and Medical Sciences, University of Bradford, Bradford BD7 1DP, UK; 2Analytical Research and Development (ARD), Pfizer Inc., Sandwich, Kent CT13 9NJ, UK

**Keywords:** EGCG, nano-liposome, inhalation, PAH, design of experiments

## Abstract

Epigallocatechin gallate (EGCG), the main ingredient in green tea, holds promise as a potential treatment for pulmonary arterial hypertension (PAH). However, EGCG has many drawbacks, including stability issues, low bioavailability, and a short half-life. Therefore, the purpose of this research was to develop and optimize an inhalable EGCG nano-liposome formulation aiming to overcome EGCG’s drawbacks by applying a design of experiments strategy. The aerodynamic behaviour of the optimum formulation was determined using the next-generation impactor (NGI), and its effects on the TGF-β pathway were determined using a cell-based reporter assay. The newly formulated inhalable EGCG liposome had an average liposome size of 105 nm, a polydispersity index (PDI) of 0.18, a zeta potential of −25.5 mV, an encapsulation efficiency of 90.5%, and a PDI after one month of 0.19. These results are in complete agreement with the predicted values of the model. Its aerodynamic properties were as follows: the mass median aerodynamic diameter (MMAD) was 4.41 µm, the fine particle fraction (FPF) was 53.46%, and the percentage of particles equal to or less than 3 µm was 34.3%. This demonstrates that the novel EGCG liposome has all the properties required to be inhalable, and it is expected to be deposited deeply in the lung. The TGFβ pathway is activated in PAH lungs, and the optimum EGCG nano-liposome inhibits TGFβ signalling in cell-based studies and thus holds promise as a potential treatment for PAH.

## 1. Introduction

Pulmonary arterial hypertension (PAH) is a rare group of life-threatening vascular disorders characterised by the abnormal production of various endothelial vasoactive mediators, for example, nitric oxide, prostacyclin, or endothelin [1]. In PAH, there is a decreased production of both nitric oxide and prostacyclin (vasodilators), and an elevated production of the potent vasoconstrictor endothelin-1 [1]. In addition, we and others have identified mutations in the bone morphogenetic protein type 2 receptor (BMPR2) and SMAD genes, which result in the excessive proliferation of pulmonary artery smooth muscle cells and the attenuation of apoptosis, which contribute to the pathogenesis of PAH [2,3,4].

Although modern pharmacological drugs have enhanced the life expectancy and quality of life of PAH patients, these medications have many drawbacks, including a short half-life, instability, a lack of organ specificity, and several formulation limitations, which limit the efficacy and increase their side effects [1]. Therefore, it is essential to find new medications or formulations that solve the limitations of the currently available medications.

Green tea contains several polyphenols [5], including many catechin compounds, such as the most abundant green tea catechin, (-)-epigallocatechin gallate (EGCG) [6]. EGCG is slightly water-soluble and has relatively high lipophilicity [7,8]. The predicted and experimental log *p*-values of EGCG are 2.38 and 0.69 ± 0.08, respectively, which means a that higher concentration of EGCG will be partitioned in the lipid layer [7,8].

EGCG has been shown to have numerous biological effects, including antioxidant, anti-thrombotic, anti-platelet, anti-proliferative, anti-mutagenic, anti-inflammatory, and anti-angiogenic effects [6,9,10]. Most recently, studies have focused on examining the molecular activity of EGCG on TGF-β signalling [11,12]. They proved that EGCG inhibits the expression of α-smooth muscle cell actin via the TGF-β-Smad2/3 pathway in MRC-5 human lung fibroblast cells [11]. EGCG ameliorates pulmonary hypertension by inhibiting the rate of proliferation of pulmonary artery smooth muscle cells by controlling mitochondrial fragmentation [10,13,14].

Despite the fact that EGCG is characterised by the aforementioned biological activities and is a promising treatment option for PAH [10,13], it has stability issues in biological and aqueous media, a very short half-life, and very low bioavailability [15,16]. These drawbacks hinder its clinical usage [15]. For example, only 0.2–2% of its orally administered concentration reaches the plasma [15]. This very low bioavailability is attributed to unfavourably high gastrointestinal and liver metabolism/degradation, transporter-mediated intestinal secretion/efflux, and poor membrane permeability. With regards to the stability issues of EGCG, it is significantly degraded in biological fluids and during storage through epimerisation and auto-oxidation [17]. The rate of its epimerisation and auto-oxidation depends on many factors, such as the concentration of EGCG, the pH of the media, and the presence of oxygen or antioxidants [17]. For example, it has been shown that EGCG has poor stability when the pH is between 4 and 8, but it has good stability when the pH is less than 4 [18,19,20]. Its half-life under many cell culture conditions is less than 2 h and is even shorter in the absence of cells [20]. Its elimination half-life is about 3.9 h [21]. In addition to these limitations, EGCG lacks target-specificity [22].

Liposomes are defined as submicron-sized spherical lipid vesicles that consist of a lipid bilayer with a hydrophilic inner cavity [23]. The structural ingredients of liposomes consist of phospholipids or synthetic amphiphiles, which can be combined with sterols such as cholesterol to enhance and alter the permeability of the liposome membrane [23]. Pulmonary delivery of inhalable liposomes has distinctive benefits over other nanoparticles because their phospholipid components are the same as those present in lung surfactants [24]. Inhaled liposomes can deliver a drug systemically or locally to the lung, provide sustained release kinetics, increase the drug’s stability and permeability, reduce the systemic side effects, and enhance the patient’s compliance [24]. Proof of the success of liposomes as a drug-loaded carrier for inhalation has been provided by a liposome inhalation suspension with the brand name Arikace^®^, which has been given approval by the Food and Drug Administration (FDA) for the management of mycobacterial infections [25]. Several other inhaled drug-loaded liposomes have reached different phases of clinical studies [24]. These inhaled liposomes are designed to target various pulmonary diseases, such as lung cancer, tuberculosis, fungal infections, and pulmonary hypertension, and are also used in lung transplantation [24]. It is worth mentioning that for the inhalable particles to reach the bronchioles and alveoli, they must be in the fine particle fraction (FPF), which means they should have an aerodynamic size smaller than 5 µm, and if the deposition target is the alveolar area, as in pulmonary hypertension disease, the inhaled particles must be smaller than 3 µm [26,27].

Pulmonary drug delivery by carriers such as liposomes has several advantages over delivering the free drug by inhalation [24,28]. First, liposome-based delivery would provide a sustained release of the drug [24,28]. Second, targeting drugs selectively to the site of action in the lung would enhance the drug’s stability and permeability [28]. Finally, liposomes prevent local irritation of the drug by entrapping the free drug inside the phospholipid’s components [28].

Based on the above review, pulmonary drug delivery is required to enhance EGCG’s stability and bioavailability and to apply a drug-targeting strategy to enhance its clinical utility. Various studies have applied nano-systems to the delivery of EGCG to solve its pharmacokinetic limitations in cancer therapy. These nano-systems include liposomes, lipid nanoparticles, and gold, as well as inorganic, protein-based nanocarriers [15,29]. Several approaches have been applied to develop EGCG liposomes, including the response surface methodology, and in silico and experimental strategies [29,30,31]. Luo et al. used the response surface method to optimise an oral EGCG-loaded liposome formulation using phosphatidylcholine, cholesterol, and Tween 80. The obtained oral liposomes had high encapsulation efficiency (85.79%) [29]. However, they included the surfactant Tween 80 (polysorbate 80, 1.08 mg/mL) in their formulation [29], which was cytotoxic to human bronchial epithelial cells when tested on lung cells for the inhalation route [32]. Ethanolic EGCG liposomes were prepared for use for intratumoral injection into basal cell carcinomas using egg phosphatidylcholine and cholesterol as the lipid bilayers [33]. A phospholipid with a low-phase transition temperature was used to prepare ethanolic EGCG liposomes, and it was previously reported by other researchers that phospholipids with a high-phase transition temperature should be used for inhalable liposomes to maintain their stability during nebulisation [27,34,35]. Moreover, several studies have reported that ethanol affects an aerosol’s performance by decreasing the fine particle mass [36,37,38]. This makes the prepared injectable EGCG unsuitable for inhalation. Other studies have used in silico and experimental strategies to develop EGCG liposomes [30]. They used 1,2-dioleoyl-sn-glycero-3-phosphoethanolamine and 1-palmitoyl-2-oleoylphosphatidyl-choline in the presence of a 5:1 molar ratio of magnesium to prepare the optimum EGCG liposome [30]. However, the resulting liposome was not suitable for inhalation because a phospholipid with a very low-phase transition temperature was used to prepare it [27,34,35]. Another study applied a quality-by-design strategy to prepare an EGCG liposome for use as an antioxidant for mesenchymal stem cells of the dental follicle [31]. Their optimum EGCG liposome contained only 221.9 µg/mL of EGCG, with a relatively low encapsulation efficiency (69.2%) [31].

However, none of these EGCG liposomes was designed for the management of PAH or for the inhalation route to selectively exert its pharmacological effect on the specific site of action; in addition, some of them used surfactants or co-surfactants in their formulations.

This study aimed to optimise the development of an inhalable EGCG nano-liposome formulation with high encapsulation efficiency using high-phase transition phospholipids to maintain the stability of the prepared liposomes during nebulisation without using any surfactants or co-surfactants. A second aim was to investigate the in vitro effects of an inhalable nano-liposome formulation on the inhibition of the TGF-β pathway as a potential treatment for PAH.

To the best of our knowledge, this is the first research to optimise and prepare an inhalable EGCG nano-liposome formulation that may have experimental and clinical applications in PAH.

## 2. Methods

### 2.1. Chemicals and Solvents

1,2-Dipalmitoyl-sn-glycerol-3-phosphate-rac-(1-glycerol) (DPPG, sodium salt) and 1,2-dipalmitoyl-sn-glycero-3-phosphocholine (DPPC) were obtained from Avanti Polar Lipids Inc., (Alabaster, AL, USA). Cholesterol, (-)-epigallocatechin gallate (EGCG), methanol, chloroform, and phosphate-buffered saline (PBS) were purchased from Sigma-Aldrich Company Ltd. (Poole, UK). The methanol, chloroform, and trifluoroacetic acid were of HPLC grade, and the other reagents were of analytical grade. Ultrapure water was produced by a Milli-Q purification system (Milli-Q, resistance of 18.2 MΩ cm at 25 °C). All the other solvents and reagents were of analytic grade purity and are commercially available.

### 2.2. Preparation of Inhalable EGCG Nano-Liposome Formulations

The thin-film rehydration method, which was established in 1965 by Bangham et al. [39], was used to prepare the inhalable EGCG liposomes. EGCG and a total of 40 mg of the two included lipids and cholesterol were dissolved in 8 mL (1:1, *v*/*v*) of methanol and chloroform. The molar ratio of DPPG was kept the same (20%) for all runs. However, when the molar ratio of cholesterol was 0, 10, and 20 (Table 1), the molar ratio of DPPC was 80, 70, and 60, respectively. The amount of the EGCG was calculated for each run individually to have a final drug-to-lipid (D/L) molar ratio of 5, 8, and 11, as indicated in Table 1. The organic solvents were evaporated via a vacuum rotary evaporator at 52 °C for 8 min, and then the thin film was left under a vacuum to evaporate the residual organic solvents from the thin film. The PBS and deionized water, at a 70:30 *v*/*v* ratio, were used as a rehydration solution. The volume of the rehydration solution was measured for each run individually to have a final total lipid concentration of 5, 10, and 15 mg/mL. The rehydration solution was then added to the lipid thin film at 60 °C and mixed using a hand mixing technique to form large multilamellar vesicles. The size of the liposomes was reduced by a Probe sonicator (Sonics & Materials. Inc., Newtown, CT, USA, 500-Watt Ultrasonic Processor, model VCX 500). The amplitude was set to 22%, the time was set to 50 s, and the pulse was set to 15 s on and 20 s off.

### 2.3. High-Performance Liquid Chromatography (HPLC)

The EGCG was quantified via reverse-phase HPLC (Agilent 1100^®^, Santa Clara, CA, UAS) with a diode array detector. The mobile phase was 0.1% trifluoracetic acid in water/methanol at 70/30% *v*/*v* in an isocratic condition [40]. The column used for this was a SUPELCOSIL LC-18-T HPLC column (5 µm particle size, L × I.D., 15 cm × 4.6 mm; temperature, 25 °C). The detection wavelength was 270 nm, the flow rate was 1 mL/min, the injection volume was 50 µL, and the run time was 5 min, with a retention time of 3.4 min. A methanol:PBS buffer (50:50 *v*/*v*) was used as the diluent for all EGCG samples. The tested detection range for the stock and standard solutions of EGCG was 1–500 µg/mL. The R^2^ of the final calibration curve and the linear equation was 0.993 and y=40.071 x+110.79, respectively.

### 2.4. Implementation of DOE

Polynomial models were constructed for the optimisation process by employing a 29-run, 4-factor, 3-level Box–Behnken design using Design Expert software to develop and optimise the inhalable EGCG nano-liposome formulation. The Box–Behnken design was selected, as it demands fewer runs in the case of 3 or 4 independent variables compared with the centre composite design, and its avoids factor extremes, since the range of these factors was determined on the basis of the literature and our screening experiments to be the best acceptable range to achieve our formulation goal [41]. Four independent variables were evaluated, namely (A) the total lipid concentration (the total concentration of the two included lipids and cholesterol in the liposome solution, mg/mL), (B) the pH of the dispersion media (the rehydration solution), (C) the molar percentage of cholesterol, and (D) the D/L molar ratio, which is also known as “loading capacity” [42]. The 3 levels of each factor were represented as −1, 0, and +1, as depicted in Table 1.

The variables were selected on the basis of data compiled from a literature review [31,41,43]. For example, it was reported that the total lipid concentration affected the encapsulation efficiency in some liposomal formulations [44], and that the pH of the dispersion media in the liposome formulations affected their sizes [45]. In addition, it was reported that the percentage of cholesterol in the liposome formulations affected the liposomes’ physical stability, including the PDI [43,46]. The D/L molar ratio is considered to be a critical factor that expresses the actual capacity of the liposome to accommodate the drug. Maximising the D:L molar ratio can optimise a liposomal formulation [42].

In this research, the range of total lipid concentration (A) was 5–15 mg/mL, since the total concentration of lipids in the majority of liposome formulations in medicines falls within this range [47]. The range of pH of the dispersion media (B) was chosen to be between 3 and 6.5 because this range is suitable for the inhalation route [48]. The molar percentage of cholesterol (C) in the range of 0–20% was chosen to measure the impact of cholesterol on the stability of the formulation, the encapsulation efficiency, and other independent factors. The molar ratio of DPPG was kept the same (20%) for all the tested formulations in this design, as the presence of the negatively charged lipid, DPPG, ensured a sufficiently negative zeta potential and thus prevented agglomeration of the liposome [49,50]. When the molar ratio of cholesterol was 0, 10, and 20, the DPPC molar ratio used was 80, 70, and 60, respectively. The D/L molar ratio (D) within the 8–11 range was used, as a higher ratio was shown to formulate unstable liposomes. As was shown in our screening experiments, when the D/L molar ratio for this formulation was higher than 11, agglomeration of the liposomes occurred. The selected responses were as follows: R1, liposome size (z-average, nm); R2, polydispersity index (PDI); R3, encapsulation efficiency (%); R4, zeta potential; R5, PDI after 1 month. The centre point (CP) was run four times to measure the curvature and precision of the production process. All 29 formulations that were proposed by the Design Expert software, as shown in Table 2, were prepared to generate, evaluate, and analyse the model. Polynomial equations that described the correlation between the dependent and independent variables were obtained.

### 2.5. Confirming the Predictivity of the Model, and Preparing the Optimum Proposed Liposome Formulation

After the analysis of all responses, the Design Expert 13 software proposed the optimum nano-liposome formulation according to the required optimal conditions selected. To confirm the predictivity of the selected model, the proposed optimal nano-liposome formulation was prepared and characterised, and the values of the experimental responses were contrasted with the predictive ones.

### 2.6. Characterisation of the Inhalable EGCG Nano-Liposome Formulations

#### 2.6.1. Inhalable EGCG Nano-Liposome Formulations’ Particle Size, Dispersity, and Zeta Potential Measurements

The particle size (Z-average), dispersity (PDI), and charge (zeta potential) of the prepared nano-liposome formulations were quantified using a Zetasizer NanoZs with dynamic light scattering (DLS, Malvern Instruments, UK) at 25 °C and a scattering angle of 173°. The measurements were performed in triplicate and the average was then calculated.

#### 2.6.2. pH Measurements of the Liposome Formulations

The pH value of the prepared formulations was measured using a pH meter (Mettler-Toledo, Leicester, UK).

#### 2.6.3. Determination of the Encapsulation Efficiency

To determine the encapsulation efficiency, the free drug and total drug concentrations in the mixture were measured. The purification of EGCG liposomes from the free EGCG was achieved using an Ultracel 50 kDa centrifugal filter. It was placed in an Eppendorf vial and centrifuged at a speed of 12,000 rpm for 12 min. The amount of free EGCG was determined from the filtrate. The collected filtrate was diluted with the same amount of methanol to allow detection by HPLC, since the diluent of the sample in this research was a 50:50 *v*/*v* buffer: methanol mixture. To determine the total EGCG content in the EGCG liposomes and the free EGCG mixture, 100 µL of the mixture was mixed for 5 min with 300 µL of methanol to confirm the complete lysis of the EGCG liposomes, and then 200 µL of the buffer was added to allow detection via HPLC. The previously mentioned HPLC method was used. The following equation was applied to assess the percentage of encapsulation efficiency: Encapsulation efficiency (%)=(1−concentration of free EGCG/Total concentration of EGCG)×100 %).

#### 2.6.4. Viscosity Measurements of the Optimum EGCG Nano-Liposome Formulation 

The viscosity of the optimum EGCG liposome formulation was evaluated using a SV-10 Viscometer (Malvern Panalytical Ltd., Malvern, UK) at a temperature of 25 °C. The average viscosity and SD were determined from three measurements.

#### 2.6.5. Osmolality Measurements of the Optimum EGCG Nano-Liposome Formulation 

A 3320 Micro-Osmometer (Model 3320) (Advanced Instruments, Wimborne, UK) was used to measure the osmolality of the optimum liposome formulation. The osmolality was examined three times, and the average and SD were then determined.

#### 2.6.6. Transmission Electron Microscopy (TEM) of the Optimum EGCG Nano-Liposome Morphology 

Using fine tweezers, a TEM grid was placed on a piece of clean filter paper with the shiny side up. A drop of the optimum liposome formulation solution was dripped onto the TEM grid and allowed to sit for 5 min. The grid was then blotted using filter paper and rinsed well through repeated dipping (a total of 50 times) in three sequential changes of distilled carbonate-free water. A single drop of Uranyless was placed on a piece of parafilm, and the grid was placed on the Uranyless drop for 5 min to allow the heavy metals to provide contrast. The grid was again picked up with tweezers and rinsed well through repeated dipping (a total of 50 times) in three sequential changes of clean distilled carbonate-free water. Finally, it was dried for 5 min before being inserted into the TEM column and observed [51,52].

### 2.7. Determination of the Aerodynamic Behaviour of the Optimum EGCG Nano-Liposome Formulation Using NGI

The particle distribution of the optimum EGCG nano-liposome formulation was measured using NGI (Copley Scientific, Nottingham, UK). The NGI was cooled to 5 °C with the aid of the cooling system for at least 1.5 hours to prevent the evaporation of the aerosol droplets [53]. The NGI was enclosed, and the throat was attached. A t-piece was used to connect the nebuliser to the NGI’s throat. The nebuliser’s chamber was filled with 5 mL of the purified optimum EGCG liposome formulation, which contained 3439 µg of the EGCG, which were aerosolised by vibrating the mesh nebuliser (Aeroneb GO, Aerogen Inc., Chicago, IL, USA). The flow rate was set to 15 L/min as recommended by the USP [54].

Different volumes of methanol were used for washing and bursting all the liposomes from all stages of the NGI to allow the measurement of a quantifiable concentration: 5 mL each for the nebuliser, T-piece, throat, and NGI Stages 1 to 4 was used; for NGI Stages from 5 to 7 and the micro-orifice collector (MOC), 3 mL of methanol was used for each. The solutions were then diluted with the same amount of PBS buffer to allow detection by HPLC. Three runs were performed using the optimum EGCG liposome formulation. A 2 mL sample was taken from every single flask and filtered via 0.22 µm membrane syringe filters into the HPLC vials for analysis. The mean of the total recovered dose, the recovered dose fraction, the total delivered dose, the emitted fraction (EF), the mass median aerodynamic diameter (MMAD), the geometric standard deviation (GSD), the FPF, and the fraction of particles equal to or less than 3 µm were calculated by Copley Inhaler Testing Data Analysis Software (CITDAS) [Copley scientific limited, Nottingham, NG4 2JY, UK].

### 2.8. Stability of EGCG Nano-Liposome after Nebulisation

The liposome size, PDI, and encapsulation efficiency were assessed in triplicate after generation of the aerosol to assess the impact of nebulisation on the stability of the prepared liposome.

### 2.9. In Vitro Test of the Effectiveness of the Optimum EGCG Nano-Liposome Formulation and free EGCG on the TGF-β Pathway

Cell culture, DNA transfection, and cell-based reporter assays were carried out using established protocols [3,55]. HEK293T cells were selected in this study, as they are easy to transfect and have a high transfection efficiency. HEK293T cells were seeded in a half-area of a 96-well tissue-treated microtiter plate and incubated for 24 h at 37 °C. The cells were transfected with plasmids encoding the TGFBRII receptor gene (100 ng), the SBE-Luc reporter (100 ng), and the pJ7Lac-Z plasmid (50 ng) using GeneJammer transfection reagent (Stratagene, San Diego, CA, USA) following the manufacturer’s instructions. Cells were then treated with drugs at 0.001 µM, 0.01 µM, 1 µM, and 10 µM for 24 h. Twenty-four hours after the treatment, the cells were lysed with 1× Reporter Lysis Buffer (Promega, Madison, WI, USA). Measurement of the luciferase and b-galactosidase was carried out as described elsewhere [3].

### 2.10. Statistical Analysis

Design Expert Version 13 was used to perform the statistical analysis of the DOE. The level of significance of the regression model was analysed using the ANOVA test. The ANOVA test was assessed for each response in the quadratic model of the response surface to select the most fitted model on the basis of the F-test and the degree of significance of the model according to the *p*-value. The values with *p* < 0.05 were considered to be statistically significant. Moreover, the coefficient of determination (R^2^), adjusted R^2^, predicted (R^2^), and coefficient of variation (CV%) were applied to confirm the adequacy of the selected model. One-way ANOVA and Tukey’s post-hoc tests were used for statistical analysis of the effect of the EGCG nano-liposomes on TGFβ signalling.

## 3. Results and Discussion

### 3.1. Optimisation of the Inhalable EGCG Using DOE

Although EGCG is an excellent target as a potential treatment option for several diseases because it has many biological activities, it has very low bioavailability and stability and a short half-life [6,15,56,57,58]. Several studies have attempted to address these limitations by formulating EGCG liposomes, but these studies were found to have several limitations (see Section 1). In addition, there has been no attempt to prepare EGCG for delivery through the inhalation route, and none of the studies has targeted PAH.

Consequently, the DOE was implemented in this research by applying the response surface methodology of Box–Behnken design to develop and optimise an inhalable EGCG nano-liposome formulation using high-phase transition phospholipids, and to maintain the stability of the prepared liposome during nebulisation, without using any surfactants or co-surfactants such as Tween 80 or ethanol in the formulation as a potential treatment for PAH. Liposomes were selected as the carrier for EGCG, as their phospholipid components are the same as those present in the lung’s surfactant [24]. Moreover, they provide sustained release kinetics and increase the drug’s stability [24]. On the other hand, by developing the EGCG as an inhaled liposome, it could be delivered systemically or locally to the lung, thus reducing the systemic side effects, enhancing the patients’ compliance, increasing its permeability, and preventing it from being affected by first-pass metabolism [24]. The proposed 29 runs of Box–Behnken design were performed and their results were assessed by Design Expert 13 software (see Table 2).

All the responses fitted the reduced quadratic model, except for the last response (PDI after 1 month), which was better fitted by the reduced two-factor interaction model (2FI). Table 3 shows the minimum values, maximum values, average, and SD and the chosen model for each of the given responses. Selection of the model was based on the ANOVA test, where the suggested model was not aliased, and had a significant *p*-value, an insignificant lack of fit, and the maximum adjusted and predicted R^2^. Backward reductions of the models were applied to remove the insignificant terms in each model. All the selected models for each response had *p* < 0.0001 with an insignificant lack of fit, which meant that the selected reduced models were highly significant, and the data were properly represented by the model, respectively. The values of R^2^, adjusted R^2^, predicted R^2^, and CV% for all responses are listed in Table 4.

#### 3.1.1. The Impact of the Formulation’s Composition on the Liposomes’ Size

Liposome size is an essential characteristic in a liposome formulation. It defines the physical features and biodistribution of the liposomes [59,60]. The method of preparing the liposome affects its size, which can ranges from nanometres to several micrometres [61].

According to the vesicular arrangement, liposomes are categorised as follows. Firstly, a liposome with one lipid bilayer is known as a small unilamellar liposome with a 20–100 nm size range, and is widely produced by the sonication method [61,62]. Secondly, large unilamellar vesicles with an average size of 100 nm–1 mm can be produced through extrusion [61,62,63]. Thirdly, multilamellar vesicles with a size of more than 1 mm are usually formulated via shaking by hand [61,62].

In general, the size of liposomes used in medical applications varies from 50 nm to 450 nm [61]. This research aimed to minimize the particle size of the formulated nano-liposome to increase the mass distribution profile of the liposome’s aerosol during the nebulisation process, thus increasing the percentage of the drug that would reach the alveoli. It was reported previously by our group that by decreasing the particle size of the nebulised formulations from the microscale to the nanoscale, the output’s performance would be improved, resulting in greater lung deposition and particle distribution (higher FPF% and lower MMAD) [64].

The ANOVA test showed that the following terms have an impact on the liposomes’ size with very highly significant effects (*p* ≤ 0.0001): pH (B), molar percentage of cholesterol (C), and B^2^. Moreover, the D/L molar ratio showed a significant impact on this response with a *p*-value equal to 0.0004. However, the influence of lipid concentration on liposomes’ size was insignificant, with a *p*-value equal to 0.326.

The inverse significant correlation (*p* ≤ 0.0001) between the pH of the buffer solution and the liposomes’ size may have been caused by the protonation of the phospholipid heads, especially DPPG, in this formulation at a low pH [65]. When the protonation of DPPG heads occurs at a low pH, the electrostatic repulsion between them will increase leading to the formulation of liposomes with a larger particle size [65]. In contrast, increasing the D/L molar ratio significantly (*p* ≤ 0.0004) increased the liposomes’ size. The maximum D/L molar ratio in this design was chosen to be 11 because aggregation of the liposomes occurred when a higher D/L molar ratio was used during our screening experiment. This result agrees with the study of Brgles et al. [66], which noted that when the amount of the drug in the liposome formulation increased, the size of the liposomes increased and aggregation of the liposome occurred [66]. This explains why the minimum liposome size in this model was 85 for Run 6 when the buffer’s pH was 6.5 and the D/L ratio was 5, while the maximum liposome size value was obtained in this model when the pH was 3 and the D/L ratio was 11 in Run 5; the other factors were kept the same for both these runs. Regarding the cholesterol content in the formulation, a highly significant (*p* ≤ 0.0001) decrease in the liposomes’ size occurred when the cholesterol content increased. This finding agrees with the results obtained by Pathak et al., who investigated the effect of the cholesterol concentration on the liposomes’ size [67].

The B^2^ term also had significant negative correlations (*p* ≤ 0.0001) with the size of the liposomes, while the influence of the lipid concentration on the liposomes’ size was insignificant, with a *p*-value equal to 0.326. The following polynomial equation was obtained based on the ANOVA for the response results liposome size: $Liposome\;size=124.53-39.33\;B-21.83\;C+19.5\;D-18.5\;CD+35.97\;B^{2}.$ The effect of the factors on the liposomes’ size was represented as a 3D plot, as depicted in Figure 1.

#### 3.1.2. The Impact of the Formulation’s Composition on the Liposomes’ PDI

PDI is one of the most critical parameters in a liposome formulation. It measures the level of homogeneity of the size of the formulated liposomes [68]. To obtain homogenous particles, the PDI should be less than 0.2, while a PDI of more than 0.3 indicates the heterogeneity of the particles [40]. One of our aims in the optimisation process was to formulate an inhalable EGCG liposome with a low PDI to ensure the homogeneity of the liposome vesicles [40]. The pH and molar percentage of cholesterol proved to be the two key factors for this response, with a high antagonist influence on PDI (*p* < 0.0001), followed by B^2^ (*p* < 0.05), as was shown in the resulting polynomial equation of this response: $PDI=0.24-0.052\;B-0.093\;C+0.048\;B^{2}.$ The effect of the pH on PDI may be attributable to the protonation of the DPPG heads that alter the size of the vesicles of the formulated liposome and, consequently, the homogeneity of the liposome formulation [65]. The negative effect of cholesterol content on PDI was attributed to its stabilisation of the liposome formulation, as it can weaken the interaction between the acyl chains of the phospholipids, subsequently hindering aggregation of the vesicles, which resulted in the narrow distribution of the vesicle size of the liposome formulation (low PDI) [69,70]. The influence of the factors on the liposomes’ PDI was represented as a 3D plot, as depicted in Figure 2.

#### 3.1.3. The Impact of the Formulation’s Composition on the Encapsulation Efficiency

The lipid concentration and pH had a positive significant influence (*p* < 0.0001) on the encapsulation efficiency of EGCG liposomes. For example, the minimum encapsulation efficiency (69%) in this design was obtained by Run 14 when the pH was 3. However, the minimum and maximum percentage of encapsulation efficiency was obtained at pH = 6.5 was 88.5% and 96.5%, respectively. See Table 2. AD, BC, BD, A^2^, and C^2^ also showed significant impacts on the encapsulation efficiency (*p* < 0.05). However, the D/L molar ratio had an insignificant influence on this response (*p* = 0.4038). The following polynomial equation was obtained on the basis of ANOVA for the response results of the encapsulation efficiency: $Encapsulation\;efficiency=90.03+5.33\;A+4.19\;B-3.63\;C-6.25\;AB+3.5\;AD+3.85\;BC+4.18\;BD-3.03\;A^{2}+2.49\;C^{2}$. The effect of the factors on the liposomes’ size was represented as a 3D plot, as shown in Figure 3.

#### 3.1.4. The Impact of the Formulation’s Composition on the Liposomes’ Zeta Potential

One of the fundamental characteristics of a liposome formulation is the zeta potential. It represents the charge of the liposome vesicles and controls the aggregation or precipitation of the liposome vesicles; therefore, it alters the formulations’ stability [68].

Increasing the pH of the liposome media resulted in significantly higher absolute values of the zeta potentials (*p* < 0.0001). This trend can be explained by the fact that the deprotonated form of DPPG increases when the pH increases, causing a higher negative charge in the formulation, as was revealed previously in similar research [38]. In addition, increasing the cholesterol content (*p* = 0.0007) resulted in a significant rise in the absolute value of the zeta potential of the formulation. This can be attributed to the fact that all the formulations in this study had the same molar percentage of negatively charged DPPG; however, when we increased the cholesterol levels, we decreased the amount of the neutral lipid, DPPC. The terms B^2^, D-D/L molar ratio, and CD also had a significant impact on this response (*p* < 0.05). The following polynomial equation was obtained on the basis of the ANOVA for response results of zeta potential: $Zeta\;potential=-23.47-4.46\;B-1.96\;C+1.13\;D-3\;CD-4.26\;B^{2}$.

The effect of the factors on the liposomes’ size was represented as a 3D plot, as illustrated in Figure 4.

#### 3.1.5. The Impact of the Formulation’s Composition on the Liposomes’ PDI after One Month

Sustaining a constant size distribution of the formulated liposome (a constant PDI) for a long period of time is a demonstration of the liposomes’ stability [71,72]. The most influential factor for this response was C (molar percentage of cholesterol), which significantly antagonized this response (*p* < 0.0001). This is attributable to the fact that an increase in the cholesterol concentration results in a corresponding increase in the rigidity and stability of the liposomes, which prevents a significant change in the size and keeps the liposome suspension monodispersed [34]. The pH also impacted this response significantly (*p* = 0.0021). Increasing the buffer’s pH led to an increase in the negative zeta potential of this formulation [38]. A higher zeta potential value induces a more stable formulation, as it limits the fusion and agglomeration of the liposome vesicles, ensuring low and constant PDI values [68]. It should be mentioned that neither A (total lipid concentration) nor D (D/L molar ratio) showed a significant impact on this response individually; however, the terms AB and BD were shown to influence this response significantly (*p* < 0.05), as the obtained equation was  PDI after 1 month=0.34−0.06 B − 0.15 C + 0.073 AB − 0.09 BD. The effect of the factors on the liposomes’ size was represented as a 3D plot, as illustrated in Figure 5.

### 3.2. Preparation and Characterisation of the Optimum Proposed Nano-Liposome Formulation and Confirming the Predictivity of the Model

The following target criteria for the optimum inhalable EGCG nano-liposome were set in this research: the minimum liposomal particle size, the minimum liposomal PDI, the maximum absolute value of liposomal zeta potential (due to its negative value, in this study, it was set as a minimum in Table 5), the maximum liposomal encapsulation efficiency, the minimum liposomal PDI after 1 month, and the maximum D/L molar ratio; the target pH was set to 6, as this value is suitable for the inhalation route [48], and the other factors were set within the ranges depicted in Table 5. As indicated in Table 5, the importance was chosen to be 3 for all factors and responses, since all of them were equally important for our optimum formulation.

Design Expert software proposed some optimum solutions. The estimated optimal points for the selected solution had a lipid concentration of 10 mg/mL, a pH of 6, a cholesterol percentage of 20%, and D/L molar ratio of 11.

The suggested liposome preparation was formulated, and the experimental values of the responses were as follows: the average particle size was 105 nm, the PDI was 0.18 (see Figure 6), the zeta potential was −25.5, the encapsulation efficiency was 90.5%, and the PDI after 1 month was 0.19. The PDI of the optimum formulation was also assessed after 2 months and 3 months, and it was 0.19. Moreover, the average particle size was also assessed after 1 month, 2 months, and 3 months, and it was 107 nm, 106 nm, and 109 nm, respectively. Consequently, the optimum EGCG liposome formulation was shown to be in the nanoscale, and it was physically very stable for at least 3 months as well as having an excellent encapsulation efficiency of more than 90%. All the actual results of these responses were in strong agreement with the values predicted by the model, which demonstrated the excellent predictivity of the model.

Our next steps were to study the chemical stability and the in vitro release profile of the optimum EGCG nano-liposome to confirm the chemical integrity of the drug and phospholipids, and the release profile of the EGCG, respectively.

### 3.3. Viscosity Measurements of the Optimum EGCG Nano-Liposome Formulation 

The viscosity of the optimum EGCG liposome formulation was assessed at 25 °C using a SV-10 Viscometer (Malvern Panalytical Ltd., Malvern, UK), and it was 9 mPas.

### 3.4. Osmolality Measurements of the Optimum EGCG Nano-Liposome Formulation 

For the nebulised formulation, the range of suitable osmolality is between 130 and 500 mOsm/kg. It has been reported that an aerosol with osmolality outside this range could induce coughing and bronchoconstriction [48]. The osmolality of the optimum EGCG liposome formulation was equal to 359 ± 3 mOsm/kg, which was within the acceptable range.

### 3.5. The Morphology of the Optimum EGCG Nano-Liposome

In this study, negative staining and TEM, a well-established method for imaging liposomes, was used to image the optimum EGCG nano-liposome formulation because it is faster and simpler than cryo-TEM and requires less advanced equipment [52]. As shown in Figure 7, the optimum EGCG nano-liposome has a spherical shape and a size of around 105 nm.

### 3.6. The Aerodynamic Behaviour of the Optimum EGCG Nano-Liposome Formulation

The type of nebuliser plays a significant role in sustaining the stability of liposomes during nebulisation [27,73,74]. The vibrating mesh nebuliser has been proved to be less disruptive to liposomes’ walls [27,73,74,75]. The heat and the waves that are produced during nebulisation by ultrasonic nebulisers disrupt the lipid bilayers of the liposomes, leading to aggregation and/or drug loss [73,74]. The shearing forces generated by air-jet nebulisers may disrupt the liposomes and cause a significant loss of the encapsulated drugs [27,76,77]. Therefore, the vibrating mesh nebuliser Aeroneb^®^ GO was selected in this study for the aerosol spray of the optimum nanoliposome formulation, as it has advantages over air-jet and ultrasonic nebulisers for liposomes [27,73,74,75].

Characterisation of the lung deposition of the inhaled optimum EGCG liposome and the aerosol was conducted using the NGI (Copley Scientific, Nottingham, UK). The flow rate was set to 15 L/min in order to simulate the midpoint of tidal breathing for a healthy adult user [54].

The mass distribution profile of the aerosol of the optimum EGCG nano-liposome formulation in the NGI stages of the Aeroneb GO nebuliser at a flow rate of 15 L/m is shown in Figure 8. Its nebulisation time was 13 min. The mean of the total recovered dose was 3242.5 ± 55.8 µg, which means that the recovered dose fraction was excellent and equal to 94.3%. The mean of the total delivered dose was 2638.5 ± 50.2 µg, and a high EF was obtained that equalled 81.4%.

For the size distribution measurements, the MMAD was used, as it estimates the median size distribution of the aerodynamic particles of the aerosol. The GSD was determined to measure the droplets’ polydispersity [36]. The MMAD of the optimum EGCG liposome was 4.41 µm, with a GSD of 2.6, indicating that the emitted dose would be deposited in the lungs. FPF% was estimated to determine the percentage of liposomes that are considered to be inhalable, as they have an aerodynamic diameter of ≤5µm [36]. The FPF was 53.46%, implying that the aerosolization performance of the formulation was good. As it has been reported that particles with a MMAD of 1 to 5 µm were accumulate deeply in the lung, smaller particles are more preferable for deposition in the alveolar region [78,79]. The size distribution measurements were the same as those of the FDA-approved liposome inhalation suspension of amikacin (Arikayce^®^), which has a mass median aerodynamic diameter of 4.7 μm [80], a GSD of about 1.63, and a FPF% < 5 µm that ranged from 50.3% to 53.5% [81]. The deposition target of the aerosol droplets in pulmonary hypertension disease is the alveolar region; therefore, the inhaled particles must be smaller than 3 µm to be loaded in this deep region [27,82,83]. Consequently, the fraction of particles equal to or less than 3 µm was determined to measure the percentage of the dose that is expected to be deposited in the alveolar region, and it was 34.3% [78,79]. These in vitro results demonstrated that the prepared optimum EGCG liposome has all the properties needed to be inhalable and it is expected to be deposited in the narrower airways.

**Figure 8 pharmaceutics-15-00539-f008:**
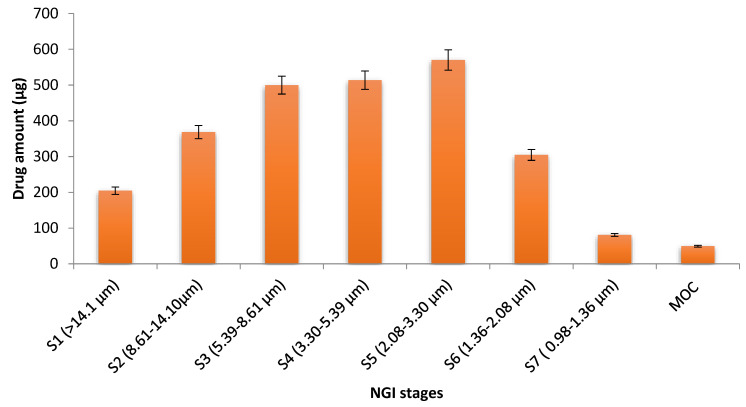
Mass distribution profile of the aerosol of the optimum EGCG liposome formulation among the NGI stages of the Aeroneb GO nebuliser at a flow rate of 15 L/min. The cut-off diameters was obtained from Marple, et al. [84].

### 3.7. Stability of the Nebulised Liposome after Nebulisation

The liposome size, PDI, and encapsulation efficiency were assessed after nebulisation to analyse the impact of the nebuliser on the stability of the prepared liposomes. Both PDI and encapsulation efficiency were not affected by nebulisation and had the same values as before nebulisation. However, the liposomes’ size rose slightly from 105 nm before nebulisation to 120 nm after nebulisation. This indicated that nebulisation does not influence the stability or membrane rigidity of the prepared liposome. This is attributed to the high-phase transition phospholipids that were used in this liposome preparation, which included DPPC and DPPG [27,34,35]. Moreover, the inclusion of cholesterol increased the rigidity of the liposomes’ wall, thus increasing the liposomes’ stability after nebulisation [34]. Another reason for the high stability of this liposome formulation after nebulisation is the type of nebuliser used in this study [73,74,75]. It has been reported that vibrating mesh nebulisers are less disruptive to the liposomes’ wall and maintain the liposomes’ stability after nebulisation [73,74,75]. The stability of our optimum EGCG nanoliposome was in strong agreement with that in the stability study on Arikace, a liposome formulation that has reached a Phase III trial [85,86]. Its success has been attributed to the use of suitable ingredients in the formulation and an appropriate inhalation device (a PARI eFlow mesh nebuliser) [85,86]. DPPC, a phospholipid with a high-transition temperature, and cholesterol were used as ingredients in the formulation [85,86]. In contrast, a significant decrease in the encapsulation efficiency was observed when ultrasonic nebulisation was applied to inhalable liposomes of sildenafil citrate, with the observed reduction in the encapsulation efficiency ranging from 12.39% to 26.23% depending on the formulation’s ingredients [87].

### 3.8. In Vitro Test of the Effectiveness of the Optimum EGCG Nano-Liposome Formulation and the Free EGCG on the TGF-β Pathway

To determine the effect of the newly formed compounds, a TGFβ-responsive reporter assay was used. The reporter assay was validated by overexpressing the TGFBRII receptor, a recognised component of the TGFβ signalling pathway (Figure 9). HEK293T cells were transfected with SBE-Luc, B-gal, and TGFBRII, for 24 h. A plasmid containing the bacterial Lac-Z gene was used as an internal standard. Cells overexpressing the receptor significantly increased the activity of the reporter, indicating the validity of the assay method. Subsequently, HEK293T cells were overexpressed with the reporter and treated with the formulated EGCG nano-liposomal compound, and their activities were compared with those of the free EGCG. Both the free EGCG and the EGCG nano-liposome formulation inhibited the reporter’s activity at a concentration of 10 µM. Interestingly, the EGCG nano-liposome inhibited the reporter’s activity at 1 µM, whilst the free EGCG failed to inhibit the reporter’s activity at this concentration. This may be attributed to the lower stability of EGCG at this lower concentration compared with 10 µM. It has been reported that EGCG’s stability depends on the concentration and that EGCG degrades quickly at low concentrations. This may also demonstrate the protective effects of the liposome formulation on EGCG’s stability [17]. The liposomal formulation itself (i.e., the lipid vehicle free from EGCG) did not elicit any discernible effect (Figure 10).

Further work is planned to confirm this in vitro effect in HEK293T cells. Cell culture validation systems will be used, including quantitative polymerase chain reaction (qPCR) and Western blotting analysis, to understand the gene expression levels. These experiments will then be further validated using monocrotaline (MCT) and hypoxia-induced PAH animal models to evaluate cell apoptosis.

## 4. Conclusions

An optimum inhalable EGCG nano-liposome was developed using high-phase transition phospholipids to maintain the liposomes’ stability during nebulisation without using any surfactants or co-surfactants in the formulation by applying the response surface methodology. Our understanding of the production method and the impact of the composition of inhalable EGCG nano-liposomes was enhanced by the implementation of the DOE strategy. The results revealed that all the studied formulation factors significantly influenced the characteristics of the prepared EGCG nano-liposome formulations. The optimum EGCG liposome formulation was stable for at least 3 months, with an encapsulation efficiency of more than 90%. The aerodynamic behaviour demonstrated the suitability of this EGCG liposome nano-formulation for inhalation and deposition in the alveolar region in PAH lungs. Moreover, the new optimum EGCG nano-liposome formulation was shown to be very stable after nebulisation with a vibrating mesh nebuliser. Whilst both the free EGCG and the optimum EGCG nano-liposomes inhibited the reporter’s activity at 10 µM, the EGCG nano-liposome formulation showed higher efficacy in inhibiting TGFβ signalling at 1 µM (*p*-value < 0.05). These points suggest that the newly formulated EGCG may have experimental and clinical applications in PAH.

## Figures and Tables

**Figure 1 pharmaceutics-15-00539-f001:**
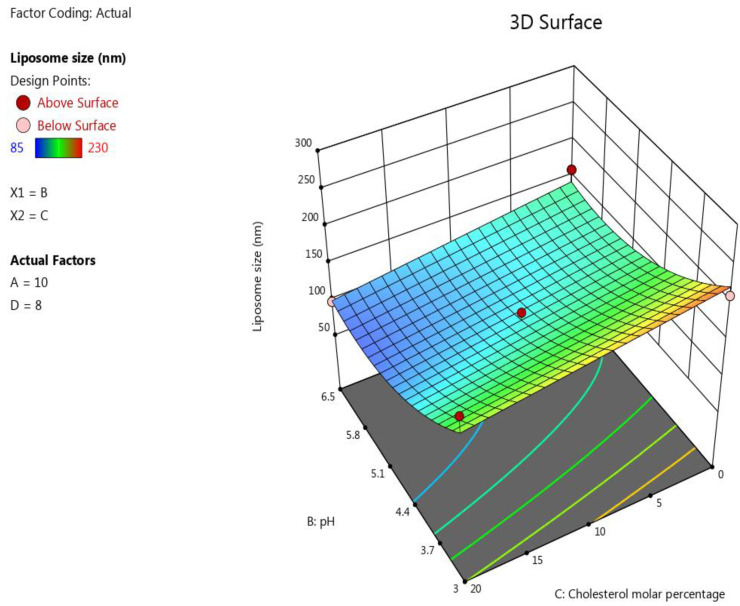
Response surface plot showing the impact of the interaction between pH (B) and the molar percentage of cholesterol (C) with a medium lipid concentration (A) and a medium D/L molar ratio (D) on the response (size).

**Figure 2 pharmaceutics-15-00539-f002:**
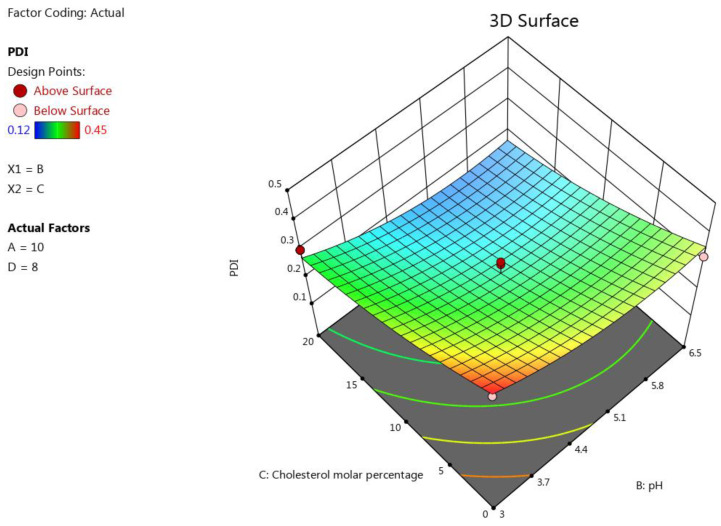
Response surface plot showing the impact of the interaction between pH (B) and the molar percentage of cholesterol (C) with a medium lipid concentration (A) and a medium D/L molar ratio (D) on the response (PDI).

**Figure 3 pharmaceutics-15-00539-f003:**
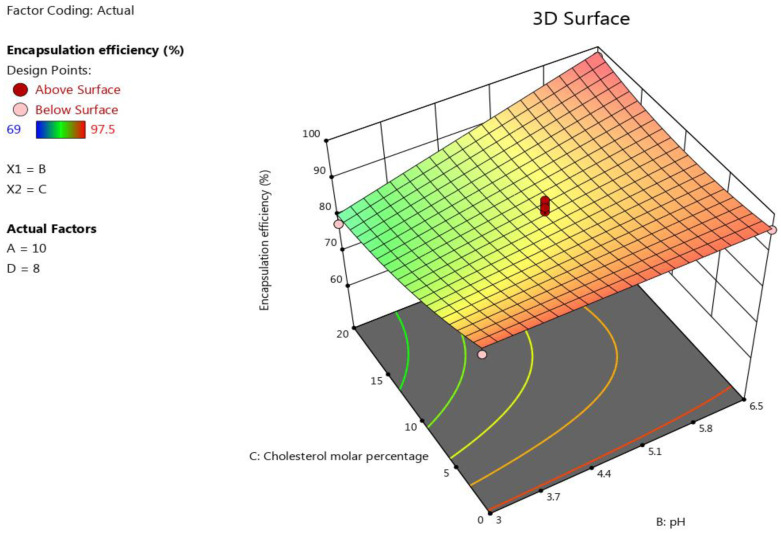
Response surface plot showing the impact of the interaction between pH (B) and the molar percentage of cholesterol (C) with a medium lipids concentration (A) and a medium D/L molar ratio (D) on the response (encapsulation efficiency).

**Figure 4 pharmaceutics-15-00539-f004:**
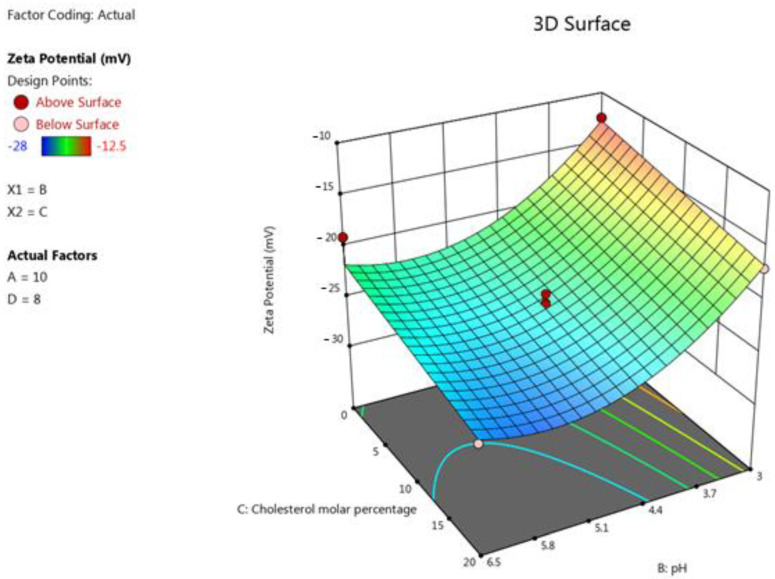
Response surface plot showing the impact of the interaction between pH (B) and the molar percentage of cholesterol (C) with a medium lipid concentration (A) and a medium D/L molar ratio (D) on the response (zeta potential).

**Figure 5 pharmaceutics-15-00539-f005:**
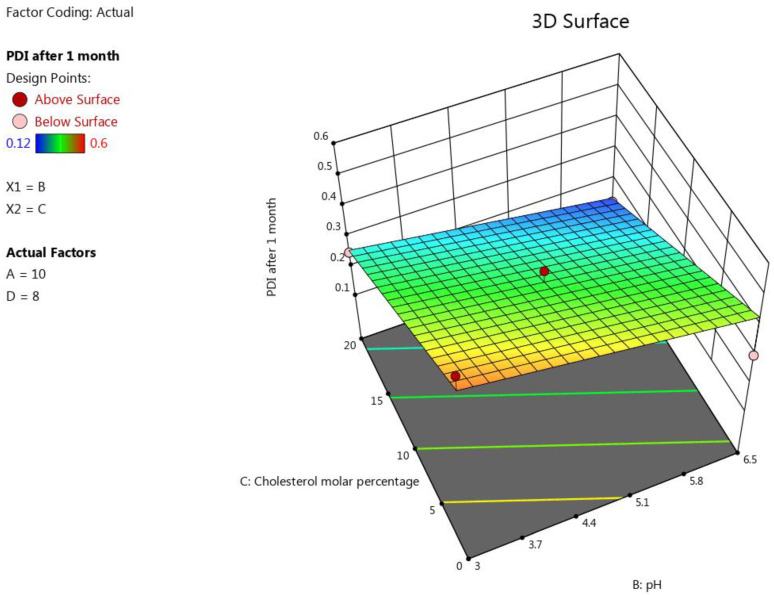
Response surface plot showing the impact of the interaction between pH (B) and the molar percentage of cholesterol (C) with a medium lipid concentration (A) and a medium D/L molar ratio (D) on the response (PDI after 1 month).

**Figure 6 pharmaceutics-15-00539-f006:**
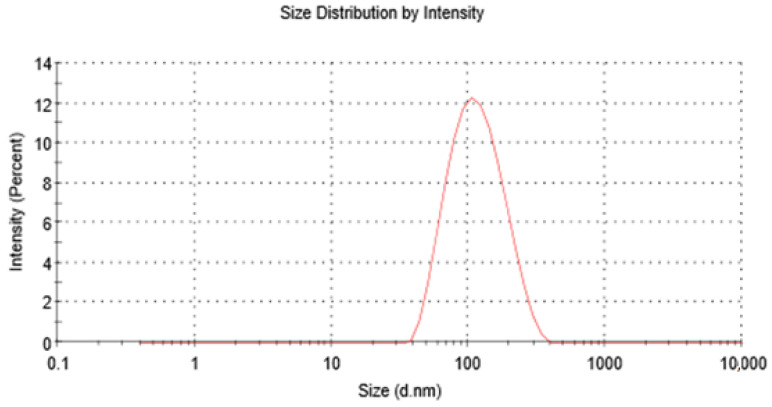
Optimum inhalable EGCG nano-liposomes’ size and size distribution.

**Figure 7 pharmaceutics-15-00539-f007:**
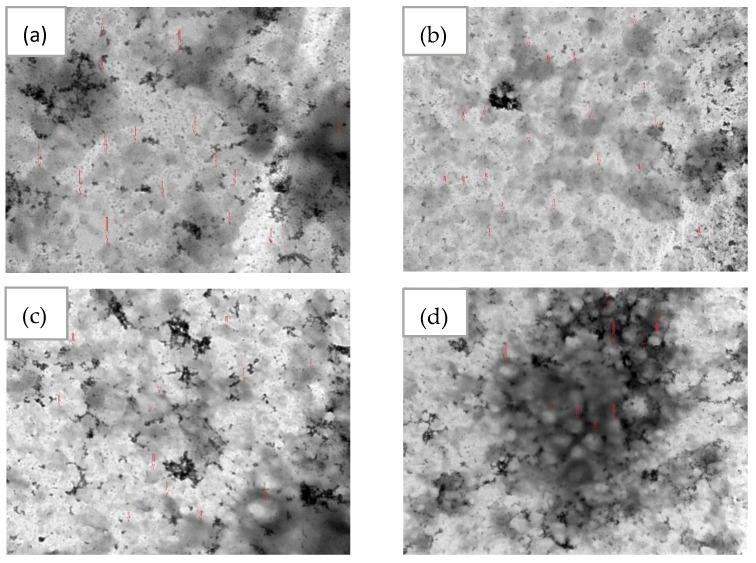
TEM images of the optimum EGCG nano-liposomes: (**a**) 80 K magnification; the scale bar represents 200 nm; (**b**) 50 K magnification; the scale bar represents 500 nm; (**c**) 40 K magnification; the scale bar represents 500 nm; (**d**) 25 K magnification; the scale bar represents 1000 nm. The red arrows show examples of liposome particles.

**Figure 9 pharmaceutics-15-00539-f009:**
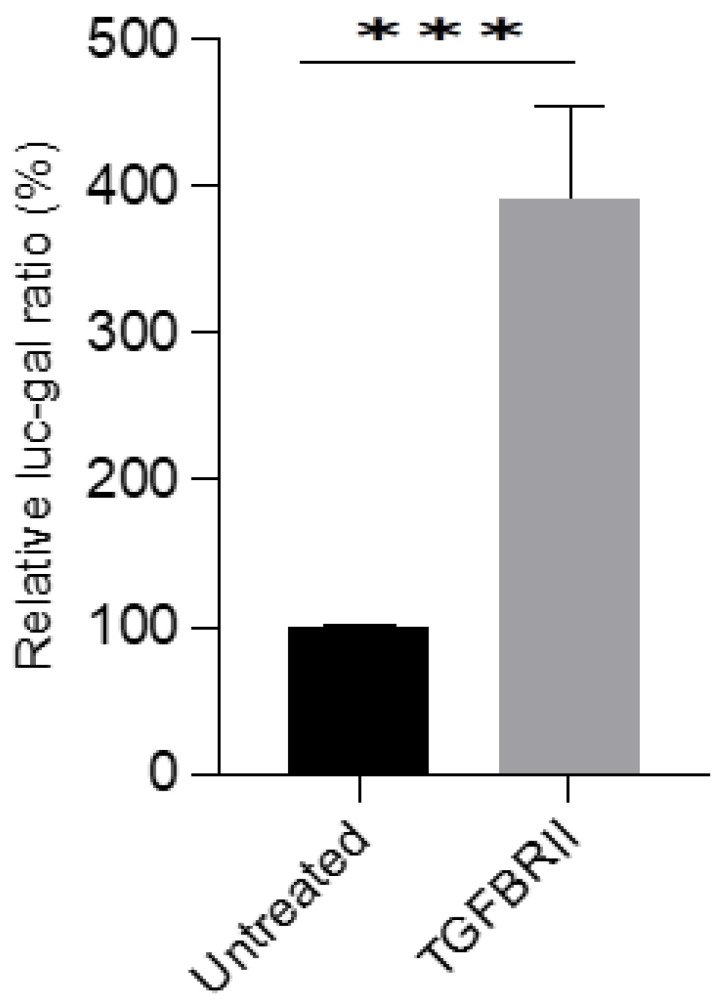
Validation of the SBE-Luc reporter assay. HEK293T cells were transfected with SBE-Luc, β-gal, and TGFBRII, for 24 h. The untreated ratio was set to 100%, cells transfected with SBE-Luc and β-gal for 24 h. Mean values of the relative Luc-Gal ratio (%) followed by the standard error of mean (SEM) were used to plot this graph in GraphPad Prism Version 9. An unpaired parametric *t*-test was used for statistical analysis, where the APA style *p*-values were used: <0.001 (***).

**Figure 10 pharmaceutics-15-00539-f010:**
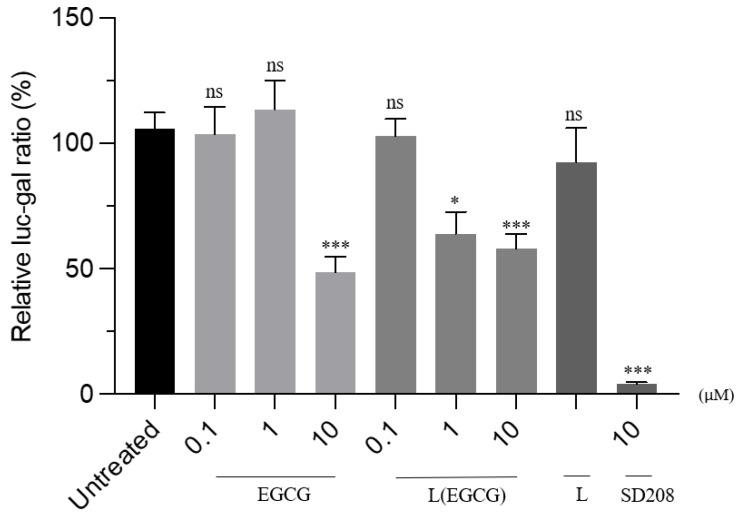
The effects of the EGCG nano-liposome formulation on TGF-β signalling. After 24 h of seeding the HEK293T cells in a 96-well half-area plate, cells were then transfected with TGFBRII, SBE-Luc, and βgal. After 24 h, these were treated with the compounds EGCG (free EGCG) and L(EGCG) (EGCG nano-liposomes at various concentrations), and L (liposomes without EGCG). These treatments were compared with the untreated TGFBRII, SBE-Luc, and βgal, which was set as 100. One-way ANOVA and Tukey’s post-hoc test were used. An unpaired parametric *t*-test was used for statistical analysis, where the APA style *p*-values were used: 0.12 (ns), 0.033 (*), and <0.001 (***).

**Table 1 pharmaceutics-15-00539-t001:** The value of the independent variables’ levels.

Independent Variables			Levels	
		Low (−1)	Mid-Point (0)	High (+1)
A	Total lipid concentration (mg/mL)	5	10	15
B	The pH of the dispersion media	3	4.75	6.5
C	Molar percentage of cholesterol (%)	0	10	20
D	D/L molar ratio (%)	5	8	11

**Table 2 pharmaceutics-15-00539-t002:** Complete Box–Behnken design. A: total lipid concentration; B: pH of the dispersion media; C: molar percentage of cholesterol; D: drug-to-lipid molar ratio; EE: encapsulation efficiency; F1: Factor 1; F2: Factor 2; F3: Factor 3; F4: Factor 4; PDI: polydispersity index; R1: Response 1; R2: Response 2; R3: Response 3; R4: Response 4; R5: Response 5.

	F1	F2	F3	F4	R1	R2	R3	R4	R5
Run	A (mg/mL)	B	C (%)	D (%)	Liposome Size (nm)± SD	PDI ± SD	EE (%)	Zeta Potential (mV)	PDI after 1 Month ± SD
1	10	6.5	20	8	98 ± 1.3	0.12 ± 0.01	95	−26	0.12 ± 0.01
2	10	4.75	0	5	94 ± 0.4	0.33 ± 0.01	96.5	−28	0.55 ± 0.12
3	10	3	20	8	200 ± 1.6	0.30 ± 0.02	78	−17	0.25 ± 0.06
4	10	4.75	20	11	102 ± 0.4	0.16 ± 0.00	87.3	−26	0.28 ± 0.01
5	10	3	10	11	230 ± 10.3	0.40 ± 0.05	80.3	−15	0.50 ± 0.05
6	10	6.5	10	5	85 ± 0.60	0.25 ± 0.01	88.5	−24	0.38 ± 0.02
7	15	4.75	10	5	105 ± 1.5	0.27 ± 0.00	86	−24	0.38 ± 0.01
8	10	3	10	5	165 ± 1.6	0.28 ± 0.01	91.5	−15	0.26 ± 0.01
9	10	6.5	10	11	144 ± 1.4	0.26 ± 0.02	94	−21.5	0.25 ± 0.02
10	15	6.5	10	8	120 ± 0.5	0.26 ± 0.00	94	−26.5	0.38 ± 0.01
11	10	4.75	10	8	135 ± 9.5	0.22 ± 0.01	90	−22.9	0.38 ± 0.01
12	10	4.75	10	8	112 ± 1.9	0.28 ± 0.03	91	−26	0.30 ± 0.03
13	5	4.75	10	5	120 ± 1.0	0.17 ± 0.05	82.5	−26	0.30 ± 0.03
14	5	3	10	8	220 ± 3.5	0.40 ± 0.03	69	−13	0.50 ± 0.03
15	10	4.75	0	11	170 ± 2.7	0.40 ± 0.05	95.5	−18	0.60 ± 0.03
16	15	3	10	8	174 ± 2.3	0.25 ± 0.02	96	−16	0.33 ± 0.02
17	5	4.75	10	11	119 ± 0.3	0.22 ± 0.01	77.5	−25	0.29 ± 0.03
18	10	6.5	0	8	160 ± 1.1	0.32 ± 0.02	96	−19	0.30 ± 0.01
19	15	4.75	10	11	138 ± 0.2	0.21 ± 0.01	95	−22	0.21 ± 0.01
20	15	4.75	0	8	167 ± 2.3	0.36 ± 0.01	95.5	−21.5	0.50 ± 0.05
21	5	4.75	0	8	180 ± 1.7	0.39 ± 0.02	93	−18.5	0.50 ± 0.02
22	15	4.75	20	8	109 ± 0.4	0.20 ± 0.01	92.5	−24	0.21 ± 0.01
23	5	4.75	20	8	110 ± 0.3	0.19 ± 0.01	81	−24	0.21 ± 0.01
24	5	6.5	10	8	120 ± 0.5	0.25 ± 0.00	92	−25	0.26 ± 0.01
25	10	4.75	20	5	100 ± 1.2	0.16 ± 0.01	93.5	−24	0.20 ± 0.02
26	10	4.75	10	8	116 ± 1.1	0.22 ± 0.01	94	−24	0.28 ± 0.02
27	10	3	0	8	210 ± 11.7	0.45 ± 0.07	94.4	−12.5	0.60 ± 0.04
28	10	4.75	10	8	118 ± 2.4	0.27 ± 0.04	93	−22	0.34 ± 0.04
29	10	4.75	10	8	122 ± 8.3	0.22 ± 0.01	92	−23	0.30 ± 0.02

**Table 3 pharmaceutics-15-00539-t003:** Minimum values, maximum values, means, SD, and selected significant models for all five dependent variables.

Response	Minimum	Maximum	Mean	SD	Model
Liposome size (nm)	85	230	139.41	40.22	Quadratic
PDI	0.12	0.45	0.2693	0.0836	Quadratic
Encapsulation efficiency (%)	69	97.5	90.03	7.08	Quadratic
Zeta potential (mV)	−28	−12.5	−21.70	4.35	Quadratic
PDI after 1 month	0.12	0.6	0.3434	0.1270	Modified 2FI

**Table 4 pharmaceutics-15-00539-t004:** Summary of the results of the regression analysis for all responses.

Responses	R^2^	Adjusted R^2^	Predicted R^2^	% CV
Liposome size (nm)	0.868	0.839	0.785	11.56
PDI	0.826	0.788	0.690	14.29
Encapsulation efficiency (%)	0.874	0.814	0.676	3.32
Zeta potential (mV)	0.875	0.841	0.767	7.97
PDI after 1 month	0.811	0.761	0.679	18.10

**Table 5 pharmaceutics-15-00539-t005:** Target criteria for the optimum inhalable EGCG liposome.

	Goal	Lower Limit	Upper Limit	Importance
A: Lipid concentration (mg/mL)	is in range	5	10	3
B: pH	is target = 6	3	6.5	3
C: Cholesterol (%)	is in range	0	20	3
D: D/L molar ratio	maximise	5	11	3
Liposome size(nm)	minimise	85	120	3
PDI	minimise	0.1	0.3	3
Encapsulation efficiency (%)	maximise	80	97.5	3
Zeta potential (mV)	minimise	−28	−12.5	3
PDI after 1 month	minimise	0.1	0.3	3

## Data Availability

Not applicable.

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
