# Peer review of "Development and Optimisation of Inhalable EGCG Nano-Liposomes as a Potential Treatment for Pulmonary Arterial Hypertension by Implementation of the Design of Experiments Approach"

_pharmaceutics, 2023, doi:10.3390/pharmaceutics15020539_

Round 1

Reviewer 1 Report

The authors have produced liposomes containing EGCG using lipids: DPPC, DPPG and cholesterol. The liposomes are intended to be delivered to the lung via nebulisation. The authors have used a Design of Experiments (DOE) approach to optimise the production of a stable formulation. The DOE approach studies the relationship between multiple input variables (factors) and key output variables (responses) in a structured manner. In addition, the authors have undertaken a limited in vitro study to investigate the effect of the liposomes vs the active ingredient alone on the TGF-β pathway.

The presentation of the study is generally clear but there are some aspects that would benefit from additional information.

Implementation of DOE. More details need to be provided on why the variables listed are chosen and how the values of the levels were chosen. The statement that they ‘were selected on the basis of data compiled from the literature review and our scereening experiments’ is not sufficient detail. It is also incorrectly stated that they are independent. PDI initially and PDI after 1 month would not generally be considered independent and total lipid and cholesterol are also not independent. The method used may assume for simplicity that they are independent but this is not the same. This should be made clear. It is also not clear what the dispersion medium is. Is this the rehydration solution mentioned in 1st para on page 5 ?

Tables and Figures. Many of them do not include all the necessary information. Frequently units are missing. Table 1 the units for the variable should be included. Table 2 Units should be included e.g. for liposome size and the meaning of A, B, C, D etc. should be made clear as notes to the Table. The meaning of F1 etc in the top row is unclear. Table 4 include units. Figures 1,2,3,4,5 replace among with between, also what does Factor Coding: Actual indicate. Table 5, it is not clear what Importance indicates in this and if all 3 is this simply indicating all equally important? Figure 7. A mass distribution profile would typically have on the x-axis indications of size ranges of the stages rather than simply a single value. Note also that the values should be µm not µM. Figure 9 this format is not appropriate for expressing significance.

The reason for using HEK293 cells is unclear.

The equation in the last paragraph on page 12 (i.e. Liposome size = ….), should be included as a formula using the correct approach for the journal

The discussion needs to be expanded to give some indication of the next steps that would be needed for confirming and further developing these liposomes for use.

Author Response

To: MDPI Singapore, Assistant Editor

Journal: Pharmaceutics (ISSN 1999-4923)

Manuscript ID: pharmaceutics-1920183

Type: Article

Title: Optimisation of the development of Inhalable EGCG nano-Liposome as a Potential Treatment for Pulmonary Arterial Hypertension by Implementation of Design of Experiments Approach

Thank you for the comments made by the referees that reviewed the above paper we submitted for publication in Pharmaceutics Journal. We have revised the paper, in accordance with the comments made, as follows

Comments:

Note: The number of pages that were inserted here are before accepting the track changes since the number of pages will be changed after accepting the track changes.

Reviewer #1:

We thank you for your comments which have been taken into consideration and addressed accordingly:

Comment: Implementation of DOE. More details need to be provided on why the variables listed are chosen and how the values of the levels were chosen. The statement that they ‘were selected considering on the basis of data compiled from the literature review and our screening experiments’ is not sufficient detail.

Response: This has been modified and all details have been added (see section 2.4, pages 6 and 7) on why the variables are chosen and how the values of the levels were chosen.

Comment:  It is also incorrectly stated that they are independent. PDI initially and PDI after 1 month would not generally be considered independent and total lipid and cholesterol are also not independent. The method used may assume for simplicity that they are independent but this is not the same. This should be made clear.

Response: Yes, we agree that PDI initially and PDI after 1 month are not independent and they are listed in this work as responses (the responses are not necessary to be independent on each other). Only the factors and not the responses are independent as has been mentioned in the manuscript. Furthermore, all four studied factors are independent. Regarding the total lipid concentration, which ranges from 5 to 15 mg/ml, it is independent from cholesterol molar percentage as the first factor (total lipid concentration) which measures the effect of the total concentration of lipid in the rehydration solution of the liposome and not the quantity of the total lipid. The cholesterol molar percentage also measures the percentage of cholesterol and not its quantity. Now, if we used the quantity (mass) of the total lipid and cholesterol as our factors, then they would be dependent on each other. For example, in the total lipid concentration, we used the same amount of lipid which was equal to 40 mg in all formulations, and we changed the volume of the rehydration solution to have different concentrations which are 5 mg/ml, 10 mg/ml and 15 mg/ml, respectively. Therefore, when we change the concentration of the total lipid, it will not affect the percentage of the cholesterol as the mass of the lipid is constant and the concentration is our factor not the mass of the total lipid.

Section 2.3 section, page 5: has been modified to include these details. (Page 5)

Comment: It is also not clear what the dispersion medium is. Is this the rehydration solution mentioned in 1st para on page 5?

Response: Yes, the dispersion medium is the rehydration solution. This has been added to the text to improve its clarity (see section 2.4, page 6).

Comment: Tables and Figures. Many of them do not include all the necessary information. Frequently units are missing. Table 1 the units for the variable should be included. Table 2 Units should be included e.g. for liposome size and the meaning of A, B, C, D etc. should be made clear as notes to the Table. The meaning of F1 etc in the top row is unclear. Table 4 include units.

Response: All the tables and figures have been modified to include all the necessary information. The units and the meaning of A, B, C, D, F1, etc have been added.

Comment: Figures 1,2,3,4,5 replace among with between, also what does Factor Coding: Actual indicate?

Response: ‘among’ has been replaced with between. Regarding the Factor coding, Design-Expert provides both the coded and the actual scale models for the user convenience. In the figures generated by the Design Expert, the actual means that the value of the factor is presented by its value and not by the coded scale. For example, the (actual coded scales) for Factor A are 5, 10, and 15 mg/ml; however, the (coded scales) for Factor A are -1, 0, 1.

Comment: Table 5, it is not clear what Importance indicates in this and if all 3 is this simply indicating all equally important?

Response: Importance in table 5 means the importance of the factors or responses in the optimizing step. In our experiment, all factors have the same importance level so yes, ‘’3’’ simply indicates that all factors and responses are equally important, and this has been added to the manuscript to make this point clear for the readers (page 21).

Comment: Figure 7. A mass distribution profile would typically have on the x-axis indications of size ranges of the stages rather than simply a single value.

Response: This has been modified (please note that Figure 7 has become Figure 8 in the revised manuscript)

Comment: Note also that the values should be µm not µM (figure7).

Response: This has been modified.

Comment: Figure 9 this format is not appropriate for expressing significance.

Response: This figure has been modified to clearly express the significance (Please note that Figure 9 has become figure 10 in the revised manuscript).

Comment: The reason for using HEK293 cells is unclear.

Response: The following clarification has been added to the manuscripts (section 2.9, page 10):

HEK293T cells were selected in this study as they are easy to transfect and have high transfection efficiency

Comment: The equation in the last paragraph on page 12 (i.e. Liposome size = ….), should be included as a formula using the correct approach for the journal

Response: All the equations in the manuscript have been modified and included as a formula using the correct approach for the journal (see section 3.1.1 page 15, section 3.1.2 page 17,  section 3.1.3 page 18,  section 3.1.4 page 19 and section 3.1.5 page 21).

Comment: The discussion needs to be expanded to give some indication of the next steps that would be needed for confirming and further developing these liposomes for use.

Response: The following texts have been added to the manuscript to clarify this comment:

  • ‘’studying the chemical stability and the in vitro release profile of the optimum EGCG nano-liposome will be our next step to confirm the chemical integrity of the drug and phospholipids, and the release profile of the EGCG, respectively’’. (Section 3.2 page 22).

  • ‘’ Further work is planned to confirm this in vitro effect in HEK293T cells, cell culture validation systems will be used including qPCR and western blotting analysis, to understand the gene expression. These experiments will then be further validated using MCT and hypoxia induced PAH animal models, to evaluate cell apoptosis. (Section 3.8 page 29).

Reviewer 2 Report

The scientific quality of the manuscript is insufficient for publication in its current form. Specific questions and points requiring attention are itemized below.

Review comments: Define “liposomes”

Review comments: It is still difficult to find the novelty of the work concerning what has already been published. What is the difference between what is published with what the authors want to publish? It is not clear. The authors must describe these differences in the introduction section.

Reviewer’s comment: Reviewer´s comment: What is the innovation of this paper? What is new in this work? It is not clear.

 Review comments: The introduction was poorly written. It requires additional information on previous attempts when similar materials were used and what were the results.

Reviewer’s comment: The results and discussion sections are poor. More comparisons with previous literature should be discussed.

Reviewer’s comment: “Liposome size is an essential characteristic in liposome formulation

as it has a significant role in liposome physical features”. The authors must indicate the liposome physical features…..Explain more in detail.

Reviewer’s comment: “.The method of liposome preparation affects its size, which ranges from nanometers to several micrometres”. What size??? Explain more in detail.

Reviewer’s comment: “This research aimed to minimize the particle size of the formulated nano-liposome to increase the drug distribution during the nebulisation process”. How is related the particle size with increasing the drug distribution???

Reviewer’s comment: Define “PDI”…

Reviewer’s comment: What is the importance of the zeta potential on the properties of liposomes?

Reviewer’s comment: The quality of the Figure 6,8 and 9 must be improved

Reviewer’s comment: Significant differences in Figure 7?

Author Response

Date: November 2022.

To: MDPI Singapore, Assistant Editor

Journal: Pharmaceutics (ISSN 1999-4923)

Manuscript ID: pharmaceutics-1920183

Type: Article

Title: Optimisation of the development of Inhalable EGCG nano-Liposome as a Potential Treatment for Pulmonary Arterial Hypertension by Implementation of Design of Experiments Approach

Thank you for the comments made by the referees that reviewed the above paper we submitted for publication in Pharmaceutics Journal. We have revised the paper, in accordance with the comments made, as follows

Comments:

Note: The number of pages that were inserted here are before accepting the track changes since the number of pages will be changed after accepting the track changes.

Reviewer #2:

We thank you for your comments which have been taken into consideration and addressed accordingly:

Comment: Define “liposomes”

Response: Definition of liposome has been added on (section 1 page 3)

Comment: It is still difficult to find the novelty of the work concerning what has already been published. What is the difference between what is published with what the authors want to publish? It is not clear. The authors must describe these differences in the introduction section.

Response: The introduction has been modified to further discuss the novelty of the work and compare what is published with what we want to publish (section 1 page 4).

Comment: Reviewer´s comment: What is the innovation of this paper? What is new in this work? It is not clear.

Response: In this research we developed and optimised an inhalable EGCG nano-liposome formulation using high-phase transition phospholipids, to maintain the stability of the prepared liposome during nebulisation, and without using any surfactants or co-surfactants such as tween 80 or ethanol in the formulation as a potential treatment for PAH with high encapsulation efficiency. However, other studies have developed EGCG liposomes, that are not suitable for inhalation as they have used low-phase transition phospholipids in their orally or injectable liposome formulations.. Moreover, other previous studies have used tween 80 in their EGCG liposome formulations. However, it has recently reported that this surfactant has potential cytotoxicity on human bronchial epithelial cells when tested on lung cell for the Inhalation route. Others have used ethanol in their prepared EGCG liposome, and it was reported by several studies that ethanol affects aerosol performance by decreasing the fine particle mass and lung deposition. Some other studies have prepared EGCG with low encapsulation efficiency. Therefore, the novelty in our formulation is that we have successfully optimised and prepared free surfactant/ co-surfactant inhalable EGCG nano-liposome using high-phase transition phospholipids. This formulation has high encapsulation efficiency and high stability during nebulisation.

This has been discussed in more details in the introduction ( page 4) and discussion (page 11) sections to address the novelty of this work.

Comment: The introduction was poorly written. It requires additional information on previous attempts when similar materials were used and what were the results.

Response: The introduction has been modified and rewritten to discuss the previous attempts with similar materials and what were the results. (Section 1.0 page 4)

Comment: The results and discussion sections are poor. More comparisons with previous literature should be discussed.

Response: More comparisons with previous literature have been added. See sections 3.6 page 27, 3.7 pages 28 and 29.

Comment: “Liposome size is an essential characteristic in liposome formulation as it has a significant role in liposome physical features”. The authors must indicate the liposome physical features…..Explain more in detail.

Response: This sentence has been modified (Page 143)

Comment: “.The method of liposome preparation affects its size, which ranges from nanometers to several micrometres”. What size??? Explain more in detail.

Response: The paragraph has been modified to include all of the required details (Page 14, section 3.1.1)

Comment: “This research aimed to minimize the particle size of the formulated nano-liposome to increase the drug distribution during the nebulisation process”. How is related the particle size with increasing the drug distribution???

Response: This sentence has been modified (Section 3.1.1 page 14, Section 3.1.1)

[67] Al Ayoub Y, Gopalan RC, Najafzadeh M, Mohammad MA, Anderson D, Paradkar A, Assi KH. Development and evaluation of nanoemulsion and microsuspension formulations of curcuminoids for lung delivery with a novel approach to understanding the aerosol performance of nanoparticles. Int J Pharm. 2019 Feb 25;557:254-263. doi: 10.1016/j.ijpharm.2018.12.042. Epub 2018 Dec 28. PMID: 30597263.

Comment: Define “PDI”…

Response: PDI definition has been added in section 3.1.2 page 16.

Comment: What is the importance of the zeta potential on the properties of liposomes?

Response: It controls the aggregation or precipitation of the liposome vesicles; therefore, it alters the formulation stability. This has been already included in the manuscript (section 3.1.4 page 19).

Comment: The quality of the Figure 6,8 and 9 must be improved

Response: The quality of these figures has been improved (please note that Figure 8 has become Figure 9 and Figure 9 has become Figure 10 in the revised version)

Comment: Significant differences in Figure 7?

Response: Figure 7 measures the aerosol mass distribution profile of the optimum EGCG liposome formulation among NGI stages. We don’t need the significant differences as we need to measure the mass distribution to calculate the mass median aerodynamic diameter (MMAD) and we didn’t compare the mass of the EGCG in each stage with other stages.

Reviewer 3 Report

The authors have focused to produce and  optimise inhalable EGCG nano-liposomes.  The formulation have been properly assessed by a QbD approach; design of experiments strategy. The science and idea for producing this project is very interesting. However, I shall suggest the following changes to be incorporated in the Manuscript before the final acceptance.

1. The Introduction Section needs to be updated with more published latest references.

2.  The manuscript  is required to be thoroughly checked for english writing mistakes.

3. Have the authors checked chemical stability of the produced liposomes.

4. In Table 2, I see the data of PDI, after one month but where is the particle size data?

5.  The particle size analysis and measurement of the PDI  for the produced samples should be done in triplicate. The SD values should be added in Table 2 against the measured PDI and P-Size values

6. I shall suggest to put SEM/TEM images of the optimised Nano Liposomes in the Manuscript.

7.  The sub sections of the the methodology need to be properly organised. For example, the method for HPLC should be put after preparation of Nano Liposomes.

8. The method section for preparation of Nano-Lipoosomes does not enfold proper informaton about quantity of different formulation ingrediends. This part needs to be updated. 

Author Response

Date: November 2022.

To: MDPI Singapore, Assistant Editor

Journal: Pharmaceutics (ISSN 1999-4923)

Manuscript ID: pharmaceutics-1920183

Type: Article

Title: Optimisation of the development of Inhalable EGCG nano-Liposome as a Potential Treatment for Pulmonary Arterial Hypertension by Implementation of Design of Experiments Approach

Thank you for the comments made by the referees that reviewed the above paper we submitted for publication in Pharmaceutics Journal. We have revised the paper, in accordance with the comments made, as follows

Comments:

Note: The number of pages that were inserted here are before accepting the track changes since the number of pages will be changed after accepting the track changes.

Reviewer #3:

Comment: The Introduction Section needs to be updated with more published latest references.

Response: The introduction section has been updated with more published latest references.

Comment: The manuscript  is required to be thoroughly checked for english writing mistakes.

Response: The manuscript has been proofread for English and it has been improved

Comment: Have the authors checked chemical stability of the produced liposomes?

Response: We studied the physical stability of the formulated liposomes. Regarding the chemical stability, we modified the discussion to include the chemical stability as our next step.

We have included the following paragraph in the manuscript ''Studying the chemical stability and the in vitro release profile of the optimum EGCG nano-liposome will be our next step to confirm the chemical integrity of the drug and phospholipids, and the release profile of the EGCG, respectively, over a period of time’’ (see section 3.2, page 21)

Comment: In Table 2, I see the data of PDI, after one month but where is the particle size data?

Response: One of the aims of this study was to examine the homogeneity of the liposome particles after one month as a response to our experimental design and also to assess the effect of the studied four factors on PDI after one month. However, for the optimum EGCG nano-liposome formulation, we studied the physical stability of this liposome, and therefore, we measured both size and PDI initially (0 time), after one month, two months, and 3 months. Section 3.2 has been modified to include these values. (Section 3.2, page 21).

Comment:  The particle size analysis and measurement of the PDI  for the produced samples should be done in triplicate. The SD values should be added in Table 2 against the measured PDI and P-Size values

Response: Table 2 has been modified to include SD values for PDI and particle size values.

Comment: I shall suggest to put SEM/TEM images of the optimised Nano Liposomes in the Manuscript.

Response: TEM images of the optimum EGCG nano-liposome have been included in the manuscript (Sections 2.6.6 page 9 and section 3.5, page 25).

Comment: The sub sections of the the methodology need to be properly organised. For example, the method for HPLC should be put after preparation of Nano Liposomes.

Response: The subsections of the methodology have been reorganised and the HPLC method has been inserted after the preparation of nano-liposome (page 5).

Comment: The method section for preparation of Nano-Lipoosomes does not enfold proper information about quantity of different formulation ingredients. This part needs to be updated. 

Response: This section has been modified to include proper information about the quantity of different formulation ingredients for all runs (Section 2.3, page 5).

Round 2

Reviewer 2 Report

The article can be accepted. However, minor comments must be attended:

1. In TEM images, the authors must point to the liposomes structure with an arrow.

Author Response

Dear Reviewer

Thank you for your suggestion for improving our manuscript. We have now included arrows in the TEM images to indicate for liposome particles as suggested
